# TNC Accelerates Hypoxia-Induced Cardiac Injury in a METTL3-Dependent Manner

**DOI:** 10.3390/genes14030591

**Published:** 2023-02-26

**Authors:** Hao Cheng, Linnan Li, Junqiang Xue, Jianying Ma, Junbo Ge

**Affiliations:** 1Department of Cardiology, Zhongshan Hospital, Fudan University, Shanghai 200032, China; 2Shanghai Institute of Cardiovascular Diseases, Shanghai 200032, China

**Keywords:** myocardial infarction, TNC, METTL3

## Abstract

Cardiac fibrosis and cardiomyocyte apoptosis are reparative processes after myocardial infarction (MI), which results in cardiac remodeling and heart failure at last. Tenascin-C (TNC) consists of four distinct domains, which is a large multimodular glycoprotein of the extracellular matrix. It is also a key regulator of proliferation and apoptosis in cardiomyocytes. As a significant m^6^A regulator, METTL3 binds m^6^A sites in mRNA to control its degradation, maturation, stabilization, and translation. Whether METTL3 regulates the occurrence and development of myocardial infarction through the m^6^A modification of TNC mRNA deserves our study. Here, we have demonstrated that overexpression of METTL3 aggravated cardiac dysfunction and cardiac fibrosis after 4 weeks after MI. Moreover, we also demonstrated that TNC resulted in cardiac fibrosis and cardiomyocyte apoptosis after MI. Mechanistically, METTL3 led to enhanced m^6^A levels of TNC mRNA and promoted TNC mRNA stability. Then, we mutated one m^6^A site “A” to “T”, and the binding ability of METTL3 was reduced. In conclusion, METTL3 is involved in cardiac fibrosis and cardiomyocyte apoptosis by increasing m^6^A levels of TNC mRNA and may be a promising target for the therapy of cardiac fibrosis after MI.

## 1. Introduction

Cardiomyocyte apoptosis and cardiac fibrosis are common features after myocardial infarction (MI). The processes lead to excessive deposition of the extracellular matrix, cardiac remodeling, and finally heart failure [1,2]. Tenascin-C (TNC), as an important extracellular matrix glycoprotein, interacts with various molecules, giving it a variety of functions during tissue injury. Due to its only absence in the normal adult myocardium, the expression of TNC reappears in the myocardium, demonstrating that the heart is under pathological conditions, like MI, dilated cardiomyopathy, or myocardial hibernation [3]. Therefore, TNC can be regarded as a key biomarker in cardiovascular diseases. Recently, TNC has been demonstrated to promote cardiomyocyte apoptosis and cardiac fibrosis, and finally cardiac remodeling and heart failure in animal models [4,5,6,7]. The inhibition of TNC has been demonstrated to attenuate cardiac fibrosis and cardiomyocyte apoptosis after MI [8,9,10], suggesting the involvement of TNC in the pathology of cardiac dysfunction after MI. 

Methyltransferase-like protein 3 (METTL3) belongs to RNA modification methylases, which regulates the expressions of mRNA or tRNA, such as the stability, degradation, and translation [11,12,13,14]. Moreover, METTL3′s biological effects are regulated by various physiological stimuli like growing factors and so on [15,16,17,18]. Therefore, the overexpression or silence of METTL3 leads to the development of many diseases [19,20,21]. Recent works have shown that METTL3 can be regarded as a potential biomarker in diagnosis or treatment in cardiovascular diseases [16,22,23,24]. It is involved in adverse cardiac remodeling after pressure overload, ischemia/reperfusion injury, and MI [24,25]. METTL3 is constitutively active and its activity is largely regulated by the expression levels in the development of cardiovascular diseases. As reported previously, the expression of METTL3 is upregulated after MI in animal models, and METTL3 silencing attenuates impaired left ventricular function and adverse cardiac remodeling [19,26]. As reported previously, the expression of METTL3 is upregulated after MI in animal models, and METTL3 silencing attenuates impaired left ventricular function and adverse cardiac remodeling [19,27]. Accumulating data have shown METTL3′s negative effects on cardiovascular diseases [19,23,27,28]. However, it has not been determined whether high expression of TNC is a cause or a result of MI. Whether METTL3 is able to trigger the m^6^A process of *TNC* mRNA directly, however, is undermined.

In the present study, we report that METTL3, an important RNA methylation enzyme, participates in the methylation process of myocardial infarction. Myocardial infarction affects the expression of METTL3, and METTL3 also regulates the apoptosis of cardiomyocytes during myocardial infarction. During in vivo experiments, we confirmed that overexpression of METTL3 promoted the deterioration of cardiac function after myocardial infarction in mice. More importantly, we found that METTL3 can participate in the methylation regulation of *TNC* mRNA. The main mechanism was that METTL3 plays a role in the m^6^A modification process by binding to the m^6^A site on *TNC* mRNA, thus promoting the stability of *TNC* mRNA. Taken together, our data suggest that the METTL3-*TNC* signaling pathway may be a novel regulation mechanism for myocardium infarction.

## 2. Materials and Methods

### 2.1. Animal Procedure and Construction of Overexpression METTL3 Model

C57/BL mice (20 g) were obtained from Shanghai JieSiJie Laboratory Animal Co., Ltd. (Shanghai, China). Mice were subjected to MI as described in our previous study [29]. Briefly, after the mice were anesthetized and ventilated with oxygen and 1.5% isoflurane (Baxter International Inc., Deerfield, IL, USA), the skin was carefully separated from the chest wall, pectoralis major, and pectoralis minor. The mosquito hemostatic forceps were used to expand the chest between the fourth rib through the gap between the pectoralis major and pectoralis minor, gently expose the location of the heart, and separate the pleura and pericardium layer by layer. Then, the anterior descending branch was ligated 2 mm below the left atrial appendage with 6–0 silk thread, and the heart was returned to its original position, Finally, 4–0 silk suture was used to suture the skin, and ECG monitoring was used to determine whether the MI model was successfully established.

METTL3 and TNC overexpression model was achieved by in situ injection of overexpression AAV9 into the heart of C57/BL mice, and the injected virus titers were 1.8 × 10^12^ PFU/mL. The stable overexpression of METTL3 and overexpression of TNC could be obtained after 3 weeks. The overexpression AAV9 was purchased from Hanheng Biotechnology Co., Ltd. (Shanghai, China). The vector was HBAAV2/9-CMV-m-METTL3-3xflag-Null and HBAAV2/9-CMV-m-TNC-3xflag-Null.

Conditional gene knockout is mainly achieved through the chromosomal site-specific recombinase system Cre-LoxP, FLP-Frt, and Dre-Rox. The most commonly used one is the Cre-LoxP system, where a LoxP sequence is placed at both ends of a target DNA sequence to be knocked out to obtain Flox mice. By mating Flox mice with tool mice expressing Cre specifically, mice with tissue or cell-specific knockout target genes can be obtained. Cre-ERT2 mice are mice that contain the fusion protein expression of the ligand binding region mutant (ERT) of estrogen receptor (ER) and Cre recombinase. Cre-ERT2 is inactive in the cytoplasm without Tamoxifen induction; when Tamoxifen is induced, the metabolite of Tamoxifen 4-OHT (estrogen analog) combines with ERT, which can make Cre-ERT2 enter the nucleus and play Cre recombinase activity. Using this method, we have constructed heart CKO-METTL3 mice. 

### 2.2. Echocardiography

The cardiac function of different groups of mice was evaluated with Vevo 770 instrument (Visual Sonics Inc., Toronto, ON, Canada) under mild anesthesia. The mice were anesthetized with inhaled isoflurane (4.5% for induction of anesthesia and 1.5% for maintenance of anesthesia). The general process was as follows: after depilation and anesthesia, the mice were placed on the heating plate in advance, sufficient ultrasonic couplant was applied to the chest and abdomen, the long axis section near the sternum was collected in the precordial area of the mice with M400 probe, and the B-section image and video are recorded. Then, the parasternal left ventricular long axis section was taken, and the M-section images were obtained at the level of mitral chordae tendineae. Finally, the Teichholz formula was used to calculate the corresponding indexes, such as left ventricular ejection fraction (LVEF: the ratio of the left ventricular systolic pump bleeding volume to the left ventricular overall volume is given to describe the left ventricular pumping function) and left ventricular shortening fraction (LVFS: left ventricular shortening fraction refers to the shortening of the left ventricle mainly from the short axis direction. Measuring the shortening rate in the short axis direction can estimate the left ventricular systolic function).

### 2.3. Histological and Immunohistochemistry Analysis

Fresh mouse heart tissue was fixed with 4% paraformaldehyde for more than 24 h and stored at 4 °C. The pathological sections were prepared by dehydration, embedding, sectioning, and other steps. After slicing, Hematoxylin-eosin (HE) and Masson staining were performed for microscopic examination and photographing [30,31]. Masson staining procedure was as followed: the slices were heated in a 60 °C oven for 15 min, and then dewaxed in xylene for 10 min; then hydrated for 5 min each time, followed by Weigert iron hematoxylin staining for 10 min and water washing for 2 min; next, acid ethanol solution was added to differentiate for 15 s, with water washing for 2 min, and Masson dye solution was added to reverse blue for 5 min, with water washing again for 2 min. Fuchsin was dyed for 10 min, and then placed in weak acid working solution for 1 min, and it was placed in phosphomolybdic acid for 2 min and weak acid working solution for 1 min again. A 95% ethanol solution was added to dehydrate for 5 s. Finally, all slices were placed in absolute ethanol dehydrate 3 times for 10 s each time, transmitted 3 times in xylene for 2 min each time, and the neutral gum seals shall be placed in a fume hood for overnight drying. HE staining procedure was similar to Masson staining. Five vision fields (×200 magnification) of each section were selected randomly under an Olympus BX-51 light microscope (Olympus, Tokyo, Japan). The percentage of microinfarct size of all samples was calculated by Image J software (Version 1.50, NIH, Bethesda, MD, USA). 

### 2.4. TTC Staining

For tissue treatment for triphenyl-2,3,5—tetrazolium-chloride (TTC) staining, the heart tissues were collected and washed with physiological saline for 10 min before being frozen at −80 °C for 20 min. Then the hearts were transferred to a heart slice mold and cut into pieces. The pieces were placed in 1% TTC at 37 °C for 30 min. Then those pieces were flipped once every 5 min and washed 3 times with ddH_2_O. All images were collected at last for further analysis. The infarct areas were measured blindly using Image software (National Institute of Health, Kanagawa, Kawasaki, Japan).

### 2.5. TUNEL Staining

One Step TUNEL Apoptosis Assay Kit (Beyotime Institute of Biotechnology, Shanghai, China) was used to identify cardiomyocytes and myocardial tissue apoptosis according to the manufacturer’s instructions. Shortly, DAPI and TUNEL methods were used to detect the apoptosis of myocardial cells. The myocardial cell nucleus was blue, and the apoptotic cell nucleus is red. The myocardium was green. The apoptotic rate was shown as the percentage of apoptotic cells in the total number of cells.

### 2.6. Cell Culture and Treatment

HL1 and AC16 cells were purchased from the Cell Bank of Type Culture Collection of Chinese Academy of Sciences. HL1 and AC16 cells were cultured with DMEM supplemented with 10% fetal bovine serum (FBS). To imitate the ischemic microenvironment, cells were exposed to hypoxia conditions at different time points. Actinomycin D (a transcription inhibitor) was used to demonstrate the stability of mRNA. 

HL1 and AC16 cells were transfected with METTL3 shRNA (Hanbio Biotechnology Co., Ltd., Shanghai, China) using Lipofectamine 3000 (Thermo Scientific, Carlsbad, CA, USA) in accordance with the manufacturer’s protocol. To decrease the expression of METTL3, HL1, and AC16 were transfected with shRNA targeting METTL3 (sh-METTL3) or non-targeting control shRNA complexed with Lipofectamine3000 Transfection Reagent at 50 nM final concentration. After 48 h, PCR was used to demonstrate the reduction level of METTL3 in each group. The virus titer was 1.7 × 10^11^ PFU/mL. The primer sequence of shRNA-METTL3: CAAGTATGTTCACTATGAA. 

### 2.7. Identification and Analysis of Methylation Sites

We first used a sequence-based N6-methyladenosine (m^6^A) modification site predictor (SRAMP) (http://www.cuilab.cn/sramp, accessed on 29 October 2022) to identify the m^6^A modification sites of TNC mRNA in mice and humans [32,33]. SRAMP is used to predict the m6A modification site on the target RNA sequence and is also used to predict the m6A modification site in mammals. SRAMP only needs RNA sequence to run the prediction and does not need to load external histological data.

### 2.8. Total RNA Extraction

Total RNA was extracted using UNIQ-10 Column Trizol Total RNA Isolation Kit (Sangon Biotech, Shanghai, China). Briefly, Trizol was added directly to the culture plate. The cracked sample was placed at room temperature for 10 min to completely separate the nucleoprotein and nucleic acid, then 0.2 mL chloroform was added, shaken vigorously for 30 s, and placed at room temperature for 3 min. All samples were centrifuged at 12,000 rpm at 4 °C for 10 min. The upper water phase was transferred to a clean centrifuge tube, 1/2 volume absolute ethanol was added and mixed well. We then put the adsorption column into the collection tube, used a pipette to add all the solution and translucent fibrous suspension into the adsorption column, left it for 2 min, centrifuged at 12,000 rpm for 3 min, and then poured out the waste liquid into the collection tube. The adsorption column was put into the recovery header, then 500 µL RPE Solution was added, centrifuged at 10,000 rpm for 30 s, and the waste liquid in the collection tube was poured out. The adsorption column was put into the recovery header and centrifuged at 10,000 rpm for 2 min. Finally, the adsorption column was put into a clean 1.5 mL centrifuge tube, 30 µL DEPC-ddH_2_O was added in the center of the adsorption membrane, then centrifuged at 12,000 rpm for 2 min, and the total RNA solution was obtained.

### 2.9. PolyA RNA Concentration

PolyA RNA was extracted from total RNA by oligo-dT affinity chromatography (NucleoTrap mRNA Kit, Macherey-Nagel, Allentown, PA, USA) following the instructions [34,35]. Briefly, 130 ug of total RNA was mixed with oligo-dT latex beads and incubated at 68 °C for 5 min, then at room temperature for 12 min. After being centrifugated at 2000× *g* and 11,000× *g*, the pallets were washed three times on the microfilter and dried by centrifugation at 11,000× *g* for 1 min. At last, polyA RNA was incubated with DEPC-ddH_2_O for 7 min at 68 °C, then centrifugated for 1 min at 11,000× *g*. The polyA RNA was purified using spectrophotometry (NanoDrop, Thermo Scientific, Waltham, MA, USA).

### 2.10. LC-MS/MS for Determination of the m^6^A/A Ratio

The polyA mRNA was digested by nuclease P1 (1 U, Sigma, St. Louis, MO, USA) in 25 mL of buffer with 20 mM NH4OAc (pH = 5.3) at 37 °C for 1 h, followed by an additional incubation with the addition of freshly made NH4HCO3 (1 M, 3 mL) and alkaline phosphatase (1 U, Sigma) at 37 °C for 4 h. All samples were diluted to 50 mL and filtered (4 mm diameter, 0.22 µm pore size, Millipore, Middlesex County, MA, USA), and 5 mL of the solution was chosen for LC-MS/MS. Nucleosides were separated by reversed-phase ultra-performance liquid chromatography on a C18 column with online mass spectrometry detection by an Agilent 6410 QQQ triple-quadrupole LC mass spectrometer in positive electrospray ionization mode. The nucleosides were quantified by using retention time and nucleoside to base ion mass transitions of 282.1 to 150.1 (m^6^A), 268 to 136 (A), 284 to 152 (G), 245 to 113.1 (U), and 244 to 112 (C). Quantification was performed in comparison with the standard curve obtained from pure nucleoside standards run with the same batch of samples. The m^6^A level was calculated as the ratio of m^6^A to A based on the calibrated concentrations [36,37].

### 2.11. SELECT for Determination of the m^6^A%

We used SELECT kit (Guangzhou Epibiotek Co., Ltd., Guangzhou, China) to determine the m^6^A sites of *TNC* mRNA bound by METTL3. The detailed protocol was following the instructions [38]. 

### 2.12. Luciferase Reporter Assay

The luciferase reporter gene is a reporting system that uses luciferin as the substrate to detect the luciferase activity of fireflies. Luciferase can catalyze the oxidation of luciferin to oxyluciferin. During the oxidation of luciferin, it will emit biological fluorescence. Then, the biofluorescence released during the oxidation of luciferin can be measured by a fluorescence meter, also known as chemiluminescence meter or liquid scintillation meter. The bioluminescence system of luciferin and luciferase can detect the expression of genes very sensitively and efficiently. For the luciferase reporter assay, AC16 or HL1 cells were transfected with indicated constructs using Lipofectamine 3000 (Thermo Scientific, Waltham, MA, USA). Then the luciferase activity was measured with an illuminometer and a Dual-Glo Luciferase Assay System (Promega, Madison, WI, USA). 

### 2.13. Statistical Analysis

All data were expressed as mean ± SEM. Two-tailed Student’s *t*-test was used to compare two different groups. For multiple groups comparison, two-way ANOVA was used. Post-hoc analysis was performed on the ANOVA using Holm–Sidak multiple comparisons or by controlling the false discovery rate using the method of Benjamini, Yekutieli, and Krieger. *p* < 0.05 was regarded as significant. All data were analyzed by GraphPad Prism version 9.0 (GraphPad Prism Software, La Jolla, CA, USA).

## 3. Results

### 3.1. TNC Promotes Cardiac Dysfunction and Cardiac Fibrosis after Myocardial Infarction

To demonstrate the effect of TNC on cardiac function, we injected OE-*TNC*-AAV9 into the myocardium before we established the MI model. After 3 weeks, we established the animal model. Next, TUNEL staining showed more apoptosis cardiomyocytes in OE-*TNC*-AAV9-injected mice after MI 1 week (Figure 1a,b). The echocardiography was used to analyze LVEF and LVIDs in each group (Figure 1c). The data showed a significant reduction in LVEF and FS% in OE-*TNC* mice (Figure 1d,e), with an increase in LVIDd and LVIDs (Figure 1f,g). Moreover, Masson’s Trichrome staining showed more fibrosis in OE-*TNC* mice (Figure 1h,i). The HE staining showed that cardiomyocytes were arranged in an orderly manner, with rich and even cytoplasm and normal stroma in the WT group, compared with the OE-*TNC* group (Figure 1h). The results showed TNC promoted cardiac dysfunction and cardiac fibrosis after MI. 

### 3.2. METTL3 Overexpression Promotes the Deterioration of Cardiac Function

In order to determine the effect of METTL3 on cardiac function, we also injected OE-*METTL3*-AAV9 into the myocardium before we conducted the MI model, after MI 3 weeks (Figure 2a,b). Next, we investigated the apoptosis levels of OE-*METTL3* and WT after MI 1 week, and we observed an increased level of apoptosis in TUNEL (Figure 2c,d). We also used TTC staining to analyze the infarct size. TTC is a mitochondrial proton pump coupling agent, which can be coupled with the mitochondrial proton pump of fresh myocardial cells for staining. The red area is the living myocardial tissue, and the white area is the infarcted myocardial tissue. The TTC staining demonstrated that more infarct size, circled by black dotted lines, in OE-*METTL3* (Figure 2e,f). Then the echocardiography was used to analyze LVEF and LVIDs in OE-*METTL3* and WT mice (Figure 2g). It showed a significant reduction in LVEF and FS% in OE-*METTL3* mice (Figure 2h,i), with an increase in LVIDs and LVIDs (Figure 2j,k). Moreover, Masson’s Trichrome staining showed more fibrosis in OE-*METTL3* mice and HE staining showed that cardiomyocytes were also arranged orderly in the OE-*METTL3* group, compared with the WT group (Figure 2l,m). It showed that METTL3 led to the deterioration of cardiac function. 

### 3.3. METTL3 Deficiency Attenuates Cardiac Dysfunction and Cardiac Fibrosis after Myocardial Infarction

In order to demonstrate the effect of METTL3 when absent, we constructed mice with cardiac METTL3-specific knockout (Figure 3a,b). We examined the apoptosis level and found less TUNEL marker compared with WT mice after 1 week of MI (Figure 3c,d). Similarly, the TTC staining demonstrated less infarct size when METTL3 was knockout, which was circled by black dotted lines (Figure 3e,f). The effect of METTL3 on LV pump function and geometry was analyzed by echocardiography in WT and CKO-*METTL3* mice 4 weeks after MI (Figure 3g). CKO-*METTL3* mice showed less reduction in LVEF after MI, compared with WT mice (Figure 3h,i). What’s more, CKO-*METTL3* mice showed a significant decrease in LVIDs (Figure 3j,k). Those data clearly exhibited that *METTL3* deficiency led to a marked increase in LV function, as well as the enlargement of LV. Then we analyzed whether CKO-*METTL3* affected myocardial fibrosis. From the results, Masson’s Trichrome staining demonstrated more fibrosis in WT mice compared with CKO-*METLL3* mice after MI (Figure 3l,m), which showed that *METTL3* deficiency reduced myocardial fibrosis. Moreover, HE staining showed that cardiomyocytes were also arranged disorderly in the WT group, compared with the *METTL3* deficiency group (Figure 3l). Those results suggested cardiac dysfunction and cardiac fibrosis were attenuated without *METTL3*. 

### 3.4. TNC Overexpression in CKO-METTL3 Mice greatly Contributes to the Cardiac Dysfunction and Myocardial Infarction

To demonstrate whether TNC contributed to cardiac dysfunction was regulated by METTL3, we injected OE-*TNC*-AAV9 into the myocardium of CKO-*METTL3* mice, and 3 weeks later we conducted the MI model. TUNEL staining showed that there was more apoptosis myocardial in CKO-*METTL3* mice injected with OE-*TNC*-AAV9 compared with that in CKO-*METTL3* mice without OE-*TNC*-AAV9 injection (Figure 4a,b). Then the echocardiography was again used to analyze LVEF and LVIDs in each group (Figure 4c). It showed a significant reduction in LVEF and FS% in CKO-*METTL3* mice injected with OE-*TNC*-AAV9 (Figure 4d,e), with an increase in LVIDd and LVIDs (Figure 4f,g). Moreover, Masson’s Trichrome staining showed more fibrosis in CKO-*METTL3* mice injected with OE-*TNC*-AAV9 (Figure 4h,i). Similarly, HE staining showed more disordered cardiomyocytes (Figure 4j). The results showed TNC overexpression in CKO-*METTL3* mice greatly led to cardiac dysfunction. 

### 3.5. METTL3 Regulates the m^6^A Process of TNC mRNA

Firstly, by using LC-MS/MS to examine the total m^6^A level of each group, the myocardial level in MI was higher than that of the sham group (Figure 5a). From SRAMP, a prediction website for exact m^6^A modification sites, predicted many m^6^A sites of TNC mRNA in mice and humans (Table 1 and Table 2). We chose HL1 and AC16 cells to demonstrate the potential mechanism. Next, we used sh-METTL3 to decrease the expression of METTL3 in HL1 and AC16 cells, and SELECT technology to examine the predicated m^6^A sites in TNC mRNA; the data from SELECT technology showed that the proportion of m^6^A/A in the exact site is less in HL1 and AC16 cells treated with sh-METTL3 compared with the control group (Figure 5b–d). To demonstrate whether METTL3 directly bound the exact m^6^A site, we mutated the “A” to “T” and used Luciferase Reporter to show that after mutation, the relative luciferase activity of the mutated group was higher than that of the nonmutated group (Figure 5e,f). Finally, to analyze whether METTL3 playing its role in MI regulated the stability of TNC mRNA, we added actinomycin D to analyze the possibility. Actinomycin D mainly acts on RNA and inhibits RNA synthesis. The mechanism of action is that it combines with its guanine group embedded in the double strand of DNA, inhibits the activity of DNA-dependent RNA polymerase, interferes with the transcription process of cells, and thus inhibits the synthesis of mRNA. At the same time, we simulated the myocardial infarction model in vivo by treating the HL1 and AC16 cells under hypoxia conditions (simulating ischemia and hypoxia of myocardial infarction) and normoxia conditions (control group). The hypoxia condition is 95% nitrogen plus 5% carbon dioxide. Therefore, we added Actinomycin D into the HL1 and AC16 cells and then detected the remaining TNC mRNA after 48 h. The result showed under the loss of METTL3, the TNC mRNA stability was lower than that of the presence of METTL3 (Figure 5g,h). Finally, we reexamined the TNC mRNA expression in CKO-METTL3 mice after injecting OE-TNC-AAV9, and it showed that the TNC mRNA level in CKO-METTL3 mice with TNC injection was higher than that of CKO-METTL3 without TNC overexpression (Figure 5i). To sum up, the results demonstrated that METTL3 regulated the m^6^A modification of TNC mRNA. 

## 4. Discussion

The distribution of heart failure in human cardiovascular diseases is highly focal. It is widely accepted that cardiac dysfunction is the final stage of myocardial infarction. Previous studies have demonstrated that involvement of various microRNAs (miRNAs), such as miR-93 and miR-484 in coordinating cell therapy responses of cardiac protection [39,40]. Our teams have also shown that, through modulating the maturation of miR-210 that targets *AIFM3* mRNAs, it regulates cardiomyocyte health and the progression of cardiac dysfunction, implicating the differential regulations of miRNA biogenesis in cardiomyocytes under hypoxia [29]. Epigenetic regulations, not only including miRNAs, DNA methylation, and histone modification, have also been implicated in the myocardial protection process [41,42]. Recently, m^6^A methylation of RNA has also come into prominence as a new important mechanism regulating various biological processes [43,44]. However, the involvement of m^6^A RNA epigenetics in MI has not been systematically investigated. 

In this study, we demonstrated that TNC promoted cardiomyocyte apoptosis, cardiac fibrosis, and dysfunction. Importantly, this phenomenon demonstrated that TNC was a key marker, with a highly upregulated expression in the myocardium during AMI or worse LV remodeling and long-term outcome in Dilated cardiomyopathy (DCM), which was consistent with Yokokawa and T’s results [45]. Kimura, T. also showed that TNC accelerated adverse ventricular remodeling after MI by modulating macrophage polarization; we demonstrated once again that the upregulation of TNC induces cardiac apoptosis and fibrosis [7]. However, less research focused on the upregulations of TNC. In this study, we showed an increase in m^6^A content and METTL3 expression in MI. Significantly, we constructed CKO-*METTL3* and OE-*METTL3* mice. By using the application of transgenic mice, we demonstrated that METTL3 deficiency decreased cardiac fibrosis and cardiomyocyte apoptosis, finally elevating the myocardial function. This observation was further corroborated by the findings of METTL3 overexpression’s effects on deterioration of cardiac function. 

In the m^6^A writer complex, METTL3 is the subunit to catalyze m^6^A methyltransferase. With the assistance of METTL4, binding the target mRNA, METTL3 is the main regulator of m^6^A. The specificity of the m^6^A process depends on the motives in different regions of mRNAs, such as 5′UTR, 3′UTR, or CDS. Due to its dynamic, the m^6^A action is associated with different cell state conditions. When a cell was moved from one condition to another one, METTL3 triggers the global m^6^A to determine the cells’ functions or even the fate [46]. Activation of hypoxia is an imitation of MI; we found that under the hypoxia condition, including the total m^6^A level and METTL3 expression, was triggered in HL1. Previous studies have demonstrated that METTL3 modified the m^6^A site of mRNA to regulate cell fate. Therefore, we analyzed METTL3 whether modified m^6^A sites of *TNC* mRNA. Notably, with the help of the SRAMP system, it predicted that there are eight very high m^6^A-modified sites in *TNC* mRNA from mice and five sites from humans. At present, RNA methylation detection generally depends on m^6^A-specific antibodies, but this method can only identify the hypermethylation region, and cannot be accurate for the detection of m^6^A at a single site. Compared with MeRIP-qPCR, the SELECT system can identify m^6^A sites and determine m^6^A fractions at single-base resolution, and can even confirm the specific target sites of various m^6^A-related “writers” and “erasers” [38,47]. We demonstrated that those predicted m^6^A sites were directly modified by METTL3 with the application of SELECT technology. 

m^6^A marks are imposed and read on a global scale to regulate transitions of cellular states. Among the targets of METTL3 under hypoxia, we identified *TNC* as a downstream effector regulating cardiomyocyte apoptosis and cardiac fibrosis. METTL3-mediated m^6^A modification on *TNC* results in an increase in mRNA stability. Consequently, the increased *TNC* mRNA will be translated into more protein to promote myocardial dysfunction. 

Although *TNC* mRNA have experienced METTL3-mediated modification that m^6^A took place in its mRNA, the next step is that modified mRNA will be “read” by m^6^A reader protein to promote its translation or degradation. For example, YTHDF1 also binds to the m^6^A-modified mRNA to stabilize the stability, whereas YTHDF2 binds to the m^6^A-modified mRNA to induce the degradation [48,49,50]. In this model, METTL3 has triggered RNA modification on *TNC* mRNA transcripts and orchestrates various m^6^A readers to determine the fate of *TNC* mRNA. The details of the RNA epigenetic regulations of *TNC*, especially the downstream molecular mechanisms, still need exploration.

## 5. Conclusions

In conclusion, our study demonstrates the important role of m^6^A RNA modification in *TNC* mRNA that promotes MI. We identify the m^6^A methyltransferase METTL3 as an important regulator that mediates the hypoxia-induced m^6^A modification on RNA transcripts to regulate cardiomyocyte apoptosis and cardiac fibrosis. Overexpression of METTL3 is able to promote hypoxia-induced *TNC* m^6^A modification to regulate myocardial dysfunction. Our work provides experimental evidence in support of the involvement of RNA epigenetics in the cellular response to hypoxia stress. The identification of METTL3 as a key regulator of hypoxia-induced pro-apoptosis and pro-fibrosis suggests a potential therapeutic approach for MI through targeting METTL3.

## Figures and Tables

**Figure 1 genes-14-00591-f001:**
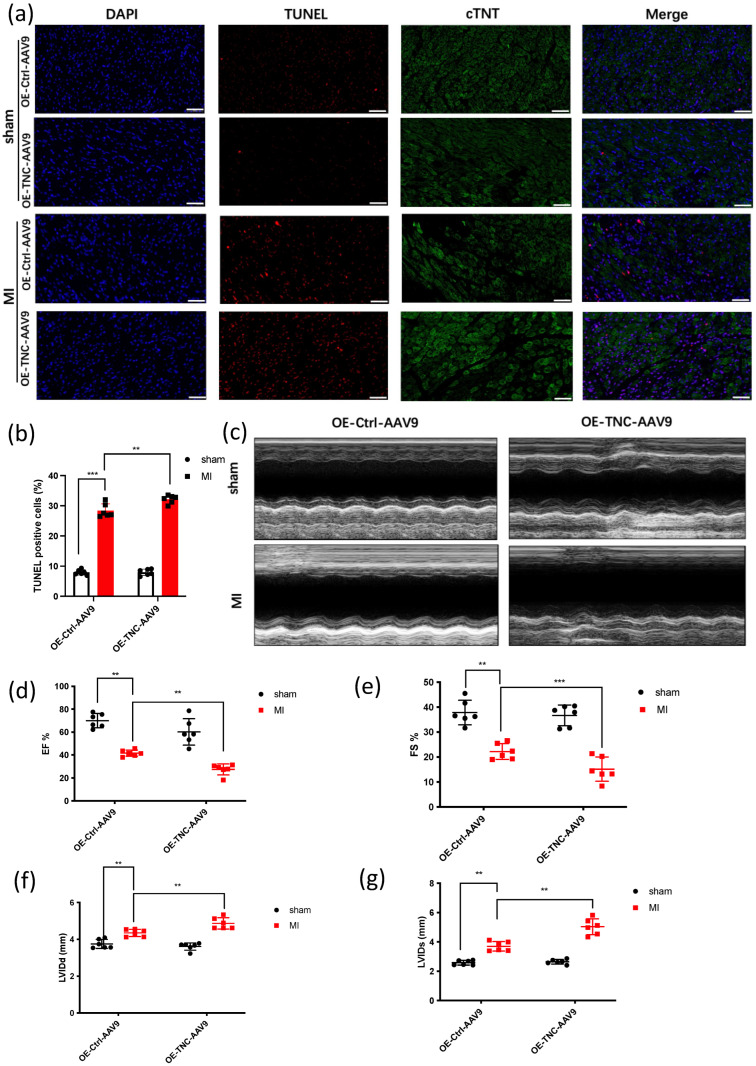
Overexpression of *TNC* leads to cardiac dysfunction and cardiac fibrosis. (**a**,**b**). Cell death in tissue samples was characterized by TUNEL assay, scar bar: 50 um, (*n* = 6); (**c**–**g**). Representative photographs of M-mode echocardiography. Quantitative analysis of echocardiography. FS, fractional shortening; EF, ejection fraction; LVIDd, left ventricular internal diameter end-diastolic: suggests that the size of the left ventricle is usually related to the impairment of cardiac function; LVIDs, left ventricular internal diameter end-systolic: reflects the size of the ventricle at the end of cardiac contraction (*n* = 6); (**h**) Masson Trichrome staining and HE staining of heart tissue sections 28 days post-ligation, scar bar: 100 um, (*n* = 6); (**i**) statistical analysis of the fibrosis in Figure 1h (*n* = 6). ** *p* < 0.01, *** *p* < 0.001.

**Figure 2 genes-14-00591-f002:**
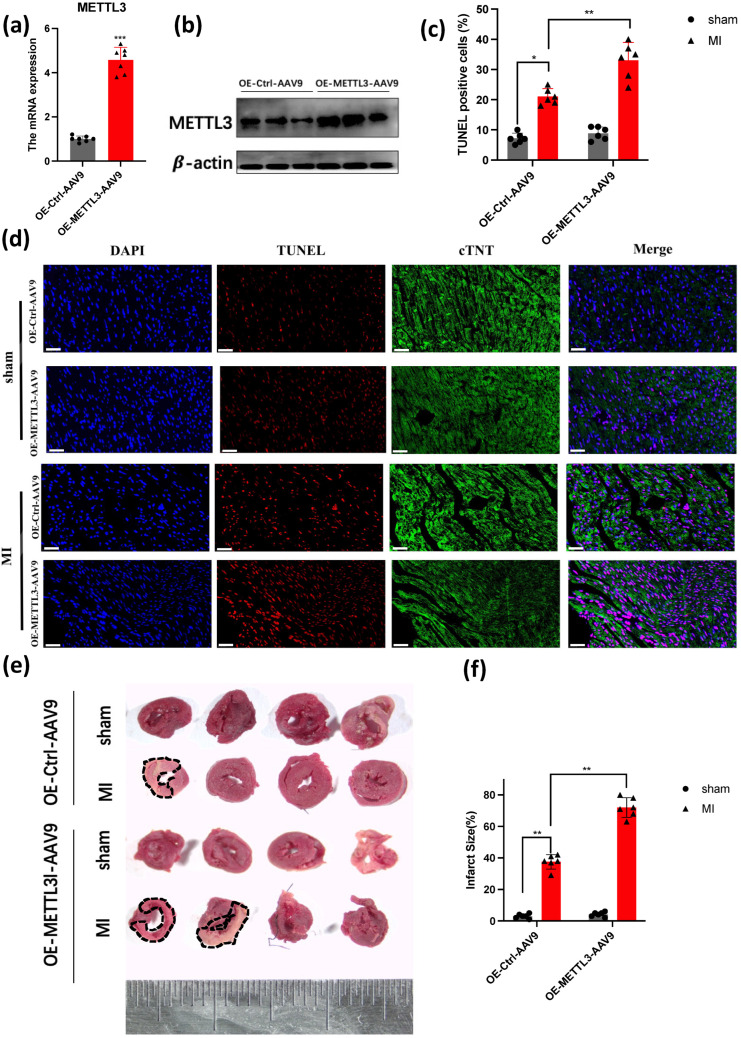
METTL3 overexpression promotes the deterioration of cardiac function. (**a**) qPCR demonstrated the overexpression of METTL3, scar bar: 50 um, (*n* = 7); (**b**) western blot demonstrated the overexpression of METTL3 (*n* = 6); (**c**,**d**) cell death in tissue samples was characterized by TUNEL assay (*n* = 6); (**e**,**f**) TTC staining characterized the myocardial injury (*n* = 6); (**g**) representative photographs of M-mode echocardiography. (**h**–**k**) Quantitative analysis of echocardiography. EF, ejection fraction; LVIDd, left ventricular internal diameter end-diastolic; FS, fractional shortening; LVIDs, left ventricular internal diameter end-systolic (*n* = 6); (**l**) Masson Trichrome staining and HE staining of heart tissue sections 28 days post-ligation. scar bar: 100 um (**m**) statistical analysis of the fibrosis in (**l**) (*n* = 6).* *p* < 0.05, ** *p* < 0.01, *** *p* < 0.001.

**Figure 3 genes-14-00591-f003:**
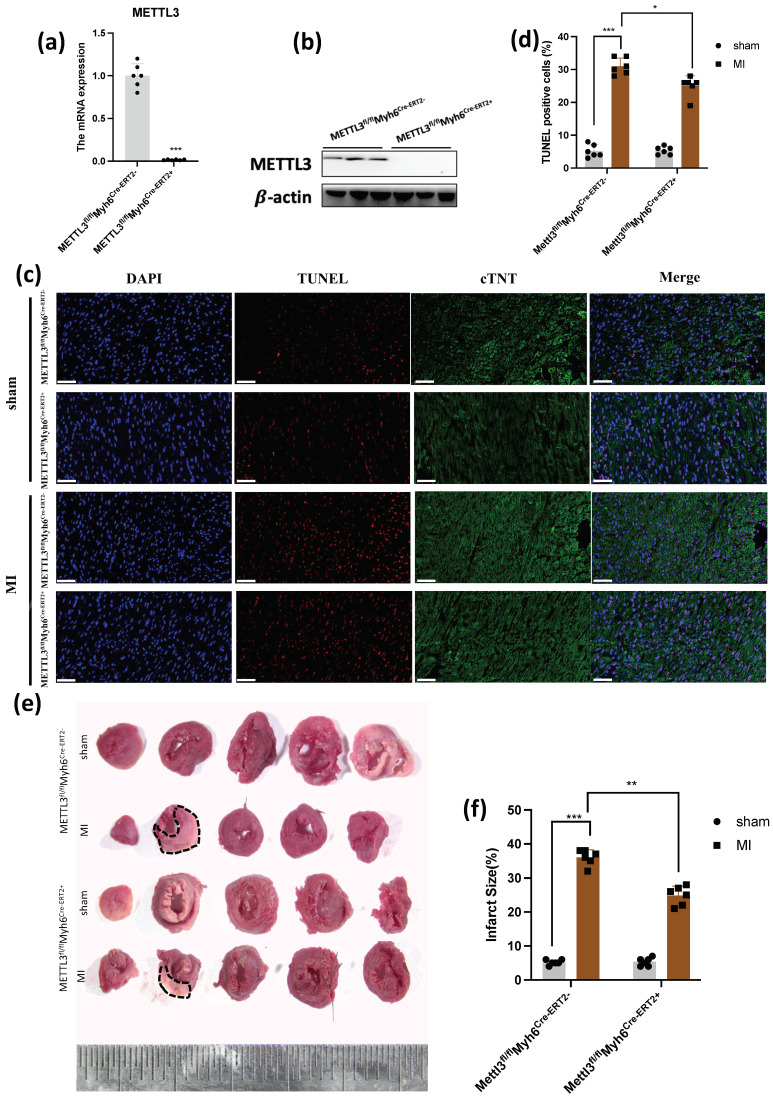
METTL3 deficiency attenuates cardiac dysfunction and cardiac fibrosis after myocardial infarction. (**a**) qPCR demonstrated the deficiency of METTL3 (*n* = 6); (**b**) western blot demonstrated the deficiency of METTL3 (*n* = 6); (**c**,**d**) cell death in tissue samples was characterized by TUNEL assay scar bar: 50 um, (*n* = 6); (**e**,**f**) TTC staining characterized the myocardial injury (*n* = 6); (**g**) representative photographs of M-mode echocardiography. (**h**–**k**) Quantitative analysis of echocardiography. FS, fractional shortening; EF, ejection fraction; LVIDd, left ventricular internal diameter end-diastolic; LVIDs, left ventricular internal diameter end-systolic (*n* = 6–15); (**l**) Masson Trichrome staining and HE staining of heart tissue sections 28 days post-ligation, scar bar: 100 um, (*n* = 6); (**m**) statistical analysis of the fibrosis in (**l**) (*n* = 6). * *p* < 0.05, ** *p* < 0.01, *** *p* < 0.001.

**Figure 4 genes-14-00591-f004:**
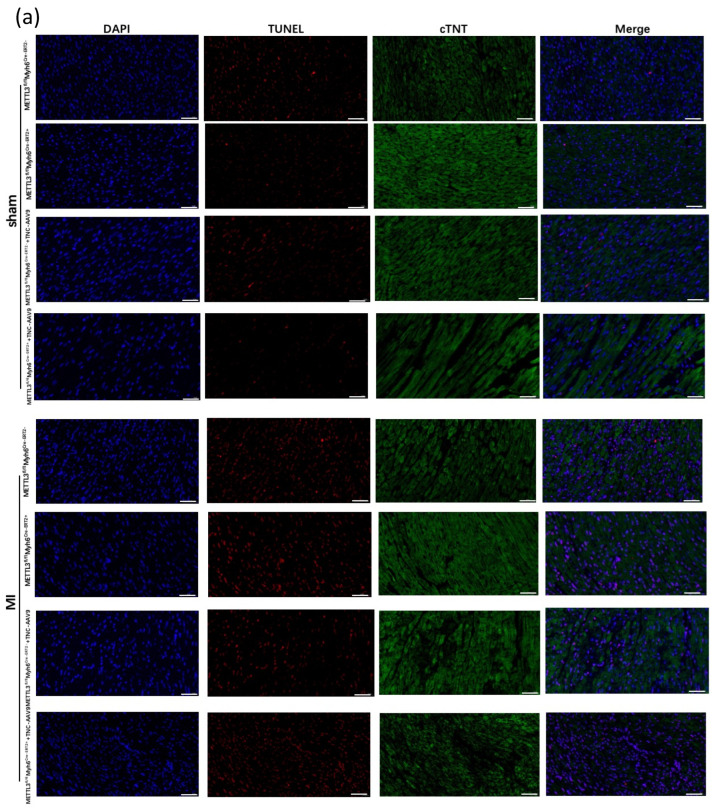
TNC overexpression in CKO-*METTL3* mice greatly contributes to the cardiac dysfunction and myocardial infarction. (**a**,**b**) cell death in tissue samples was characterized by TUNEL assay scar bar: 50 um, (*n* = 6); (**c**) representative photographs of M-mode echocardiography; (**d**–**g**) quantitative analysis of echocardiography. FS, fractional shortening; EF, ejection fraction; LVIDd, left ventricular internal diameter end-diastolic; LVIDs, left ventricular internal diameter end-systolic (*n* = 6); (**h**) Masson Trichrome staining 28 days post-ligation, scar bar: 100 um, (*n* = 6); (**i**) statistical analysis of the fibrosis in (**h**) (*n* = 6) (**j**) HE staining of heart tissue sections 28 days post-ligation, scar bar: 100 um, (*n* = 6). * *p* < 0.05, ** *p* < 0.01, *** *p* < 0.001.

**Figure 5 genes-14-00591-f005:**
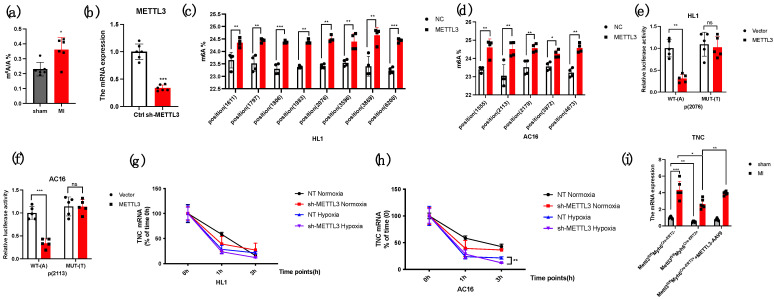
METTL3 regulates the m^6^A process of *TNC* mRNA. (**a**) LC-MS/MS was used to analyze the m^6^A/A level (*n* = 6); (**b**) the *METTL3* mRNA level after shRNA treatment (*n* = 6); (**c**) SMRAP was used to predict the m^6^A sites of *TNC* mRNA in HL1 cells and SELECT technology was used to examine (*n* = 5); (**d**) SMRAP was used to predict the m^6^A sites of *TNC* mRNA in HL1 cells and SELECT technology was used to examine (*n* = 5); (**e**) luciferase reporter was used to analyze the METTL3 directly bound to the potential m^6^A site in HL1 cells (*n* = 5); (**f**) luciferase reporter was used to analyze the METTL3 directly bound to the potential m^6^A site in AC16 cells (*n* = 5); (**g**) Actinomycin D was used to demonstrate that the stability of *TNC* mRNA was regulated by METTL3 in HL1 cells (*n* = 5); (**h**) Actinomycin D was used to demonstrate that the stability of *TNC* mRNA was regulated by METTL3 in AC16 cells (*n* = 5); (**i**) the TNC mRNA expression in CKO-*METTL3* mice after overexpressing TNC (*n* = 6). * *p* < 0.05, ** *p* < 0.01, *** *p* < 0.001.

**Table 1 genes-14-00591-t001:** The predicted m^6^A sites of TNC mRNA in mice by SRAMP.

	Position	Sequence Context	Score (Binary)	Score (Knn)	Score (Spectrum)	Score (Combined)
1	1611	GACGA GGGCU AUACC GGAGA AGACU * GUAGC CAGCG GCGAU GCCCC	0.811	0.619	0.807	0.838
2	1797	GAUGA CGACU ACACU GGGGA AGACU * GCAGA GACCG GCGCU GUCCC	0.836	0.612	0.860	0.834
3	1806	UACAC UGGGG AAGAC UGCAG AGACC * GGCGC UGUCC CCGGG ACUGU	0.798	0.263	0.858	0.795
4	1983	CACGA GGGCU UCACU GGCAA AGACU * GCAAA GAGCA AAGGU GCCCC	0.882	0.614	0.840	0.852
5	2076	CAUGA GGGCU UUACG GGCCU GGACU * GUGGG CAGCG CUCCU GUCCC	0.893	0.858	0.822	0.862
6	3596	GGAUG GCCUC AGACU CAACU GGACU * GCAGA UGACC UGGCC UAUGA	0.725	0.717	0.755	0.736
7	3869	GGAUG CCCUC ACGCU CAACU GGACU * GCUCC AGAAG GAGCC UAUAA	0.739	0.600	0.663	0.701
8	6211	GCAGA GAAGA AUUUU GGCUU GGACU * GGAUA ACCUG AGCAA AAUCA	0.762	0.824	0.622	0.709

* The red base sequence represents the m^6^A modified sequence (RRACH), where the underlined “A” represents the methylated adenine.

**Table 2 genes-14-00591-t002:** The predicted m^6^A sites of TNC mRNA in human by SRAMP.

	Position	Sequence Context	Score (Binary)	Score (Knn)	Score (Spectrum)	Score (Combined)
1	1555	AGCGA GAAGA GGUGU CCUGC UGACU * GUCAC AAUCG UGGCC GCUGU	0.846	0.540	0.803	0.814
2	2113	AAGGA GCAAA GAUGU CCCAG UGACU * GUCAU GGCCA GGGCC GCUGC	0.877	0.697	0.818	0.844
3	2179	CACGA GGGCU UCACA GGCCU GGACU * GUGGC CAGCA CUCCU GCCCC	0.754	0.773	0.568	0.681
4	3972	GGAUG CCCUC AAACU CAACU GGACU * GCUCC AGAAG GGGCC UAUGA	0.741	0.662	0.609	0.684
5	4673	AUUUU AUUGU CUACC UCUCU GGACU * UGCUC CCAGC AUCCG GACCA	0.730	0.775	0.601	0.680

* The red base sequence represents the m^6^A modified sequence (RRACH), where the underlined “A” represents the methylated adenine.

## Data Availability

Not applicable.

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
