# Peer review of "TNC Accelerates Hypoxia-Induced Cardiac Injury in a METTL3-Dependent Manner"

_genes, 2023, doi:10.3390/genes14030591_

Round 1
Reviewer 1 Report
Authors are required to add more literature references in the material and method sections.
As no reference has been cited it any of the sections.
Author Response
Thank you for your suggestion. Because some experimental methods in materials and methods are directly described, we have provided a lot of references this time. For example, in HE staining and Masson staining part, we cited Bai WW’s work as number [27] (PMID: 34950416). Moreover, in terms of m6A modification, SRAMP, a prediction of mammalian m6A sites based on sequence-derived features, has been cited in [29] (PMID: 26896799). In PolyA RNA concentration part, we cited Williams JG’s work (PMID: 487435), LC-MS/MS for m6A, we cited many popular works (PMID: 35810559; 35246985). SELECT tool for m6A site demonstration was referred by Jia Guifang’s work as number [35] (PMID: 30345651).
The newly added references have been marked in red. (lines 80, 114, 156, 178, 199, 204)
Reviewer 2 Report
Dear authors,
the study focused on the role of METTL3 inducing TNC cardiac injury.the research topic is interesting and might be of scientific importance , however the overall objective of the study does not propose such novelty in terms of experiment and new mechanistic explanation.
I suggest rewriting the abstract and paying attention to the logical order of the sentences as well as in the discussion (it is difficult to follow the read), I think that highlights the importance of the method for mA analysis is not the goal of the paper;
Here I reported some difficult sentences to understand
from 29 to 32; 46 to 47.
51 misses the reference.
The last part of the introduction from 55 to 61 needs to be rewritten, deleting the methodology used.
65 the previous study for the model needs to be cited.
Most of the graphs regarding the tunnel assay have circles in both condition MI and sham and miss the color legend. In addition, as you mentioned, TNC is already known to induce fibrosis and its inhibition attenuates it. Thus, regarding the results you obtained by over-expressing TNC, how do you explain that in CTR condition, only the overexpression of TNC does not induce fibrosis in normo-condition?
I suggest plotting the quantification analysis for the Masson and HE assays in all the cases.
222 METTL (3)
Fig 1 misses the dot plot as the other graphs (as well as graphs 2f), Fig 2c misses the analysis sign and SD. From graphs 2h to 2k, graphs are not aesthetically in line with the other. Moreover, the description of fig 2e-f-g-k do not correspond to the figures
Graph 3A needs to turn in red color and 3D miss the SD
Fig 4 The anova analysis should be performed on the three conditions. As far as I understood, regarding the wt (first red bar) I expect the Tunnel assay to be higher than KO MTTL3. Please perform the analysis and case explain this result.
Could you prove that with over-expression of the two genes, fibrosis after MI is even higher, sustaining the hypothesis of your mechanisms?
Fig5 D and E have different color.
I will suggest showing the general level of mRNA expression of TNC in vivo after MI in KO and KI model to confirm the stability of the mRNA by MTTLE3
In the discussion at phrases 338 to 342, the demonstration you got needs to be clarified since there are references; maybe it would be better to highlight that you demonstrated once again that the up-regulation of TNC induces cardiac apoptosis and fibrosis.
Lately, I suggest not reporting the figure in the discussion.
At the end, I might add a comment. Data regarding the KO or OVER expression of METTL3 in reducing or increasing cardiac fibrosis is well established in the literature by other studies. It would have been better to cite them instead of reproducing data in this case. Nevertheless, I would ask you to perform experiments such as CC3/C3 and BAX/Bcl2, as well as the Caspase activities, to gain the strength of the research.
Author Response
Thank you for your suggestions!
- from 29-32, this sentence demonstrated that TNC, this protein didn’t express in normal adult myocardium. Therefore, if TNC is expressed in adult myocardium, it demonstrated that the adult myocardium was under abnormal conditions, such as myocardium infarction, dilated cardiomyopathy and so on.
- From 46-47, “METTL3 is constitutively active”demonstrated that METTL3 is a complex when exerting biological effects, It must form a complex with a variety of proteins to be active. For example, METTL14 is an another important m6A writer, it can bind with METTL3 to form the complex to add the m6A to the targeted RNAs [1, 2].
“its activity is largely regulated by the expression levels in the development of cardiovascular diseases.”demonstrated that METTL3’s activity was regulated by its expression level. METTL3 is a very important methylation enzyme. The efficiency of enzymatic reaction is regulated by the amount of enzyme expression. At the same time, the amount of METTL3 expression will change during the progression of cardiovascular disease, so the efficiency of methylation will also show dynamic changes with the progression of disease [3].
All in all, this sentence should be supported by references. 【19,26】
- 51, we added the references. 【19,23,27,28】Accumulating data have shown METTL3’s negative effects on cardiovascular diseases. At present, overexpression METTL3 appears in most cardiovascular diseases, which promotes the development of those diseases.
- For 55-61, we rewrote this part, and deleted the methodology used.
In the present study, we report that METTL3, an important RNA methylation enzyme, participates in the methylation process of myocardial infarction. Myocardial infarction affects the expression of METTL3, and METTL3 also regulates the apoptosis of cardiomyocytes during myocardial infarction. In vivo experiments,we confirmed that overexpression of METTL3 promoted the deterioration of cardiac function after myocardial infarction in mice. More importantly, we found that METTL3 can participate in the methylation regulation of TNC mRNA. The main mechanism was that METTL3 plays a role in the m6A modification process by binding to the m6A site on TNC mRNA, thus promoting the stability of TNC mRNA. Taken together, our data suggest that the METTL3-TNC signaling pathway maybe a novel regulation mechanism for myocardium infarction.
- Line 65: Mice were subjected to MI as described in our previous study. We added the reference. 【29】
- In all TUNEL images, the color bar legend is at the lower right corner of each image. You need to zoom in to see the image clearly, because when you put it in the manuscript, the image is compressed.
- For this question: how do you explain that in CTR condition, only the overexpression of TNC does not induce fibrosis in normo-condition?
Under normal circumstances, fibrosis may also occur after overexpression of TNC, but we emphasize that hypoxia, as an important promoting factor, can aggravate cardiac fibrosis under this condition. It is not that overexpression of TNC alone will not lead to fibrosis. In addition, our CTR condition in the experiment refers to METTL3 knockout. In the h of Fig4, our two groups of conditions are sham and myocardial infarction, and METTL3 knockout and control.
- We added all quantification analysis of Masson staining (Fig 1i, 2m, 3m and 4j). HE staining is mainly used to observe cardiomyocytes with abnormal morphology, so there is little statistical analysis on this in the current research, and more visual pictures are used to show the morphological differences between the experimental group and the control group.
- Line222, we corrected the METTL3 spelling.
- In this question: Fig 1 misses the dot plot as the other graphs (as well as graphs 2f), Fig 2c misses the analysis sign and SD. From graphs 2h to 2k, graphs are not aesthetically in line with the other. Moreover, the description of fig 2e-f-g-k do not correspond to the figures
1). We corrected the Fig 2f with dot plot. 2). Fig 2c, we added the analysis sign and SD, and we added the sample quantity (n=6). 3). Fig2h to 2k, we corrected these images to match the whole figure. For example, we changed the color and changed the style. 4). We corrected the description of Fig 2e, 2f, 2g, and 2k to correspond to the figures.
- Graph 3A needs to turn in red color and 3D miss the SD
We have corrected the red color in Fig 3A, and added the SD in Fig 3D. Moreover, we changed all image style and color in Fig 3 to make consistence.
- Fig 4 The anova analysis should be performed on the three conditions. As far as I understood, regarding the wt (first red bar) I expect the Tunnel assay to be higher than KO MTTL3. Please perform the analysis and case explain this result.
In terms of this question, we corrected the statistical analysis section:
All data were expressed as meanSEM. Two-tailed Student’s t test was used to compare two different groups. For multiple groups comparison, two-way ANOVA was used. Post-hoc analysis was performed on the ANOVA using Holm-Sidak’s multiple comparisons or by controlling the False Discovery Rate using the method of Benjamini, Yekutieli and Krieger. p0.05 was regarded significant. All data were analyzed by GraphPad Prism version 9.0 (GraphPad Prism Software, La Jolla, CA, USA).
Moreover, we also added the comparison between sham with KO METTL3. Your prediction is right, the TUNEL assay was higher than KO METTL3. In Fig 3d, we have compare the sham group with KO METTL3 group in MI, so in Fig 4, we didn’t decide to compare them again. But in your opinion, we added the comparison, which demonstrated that TNC’s role in KO METTL3 condition better.
- Could you prove that with over-expression of the two genes, fibrosis after MI is even higher, sustaining the hypothesis of your mechanisms?
It is a great question. It is very easy to understand that over-expression of METTL3 and TNC could promote the fibrosis after MI better, because in MI condition, fibrosis was increased after surgeon, and METTL3, as a leading cause to deteriorate the heart function, could further aggravate the fibrosis. Finally, TNC, if overexpressed, its mRNA as the target of METTL3, could be increased a lot after m6A regulation, Therefore, if overexpressed METTL3 and TNC at the same time in MI condition, the fibrosis would be aggregated a lot than single overexpressed.
- Fig5 D and E have different color.
We have changed the color in Fig 5D and 5E, and Fig 5A.
- I will suggest showing the general level of mRNA expression of TNC in vivo after MI in KO and KI model to confirm the stability of the mRNA by MTTLE3.
To demonstrate the stability of TNC mRNA by METTL3, we should analyze the TNC mRNA level in vivo. Therefore, we provided the result as Fig 5h. We mainly did the experiment in KO METTL3 mice, after MI, we found TNC was decreased a lot. After injecting the AAV-METTL3, we found that TNC was increased a lot (Fig 5h).
- In the discussion at phrases 338 to 342, the demonstration you got needs to be clarified since there are references; maybe it would be better to highlight that you demonstrated once again that the up-regulation of TNC induces cardiac apoptosis and fibrosis.
In this part, we got the demonstration that was consistent with the references. Therefore, we should clarify it. (lines 368-372)
In this study, we demonstrated TNC promoted cardiomyocytes apoptosis, cardiac fibrosis and dysfunction (Figure 1). Importantly, this phenomenon demonstrated that TNC was a key marker, with a highly up-regulated expression in the myocardium during AMI or worse LV remodeling and long-term outcome in Dilated cardiomyopathy (DCM), which was consistent with the Yokokawa, T’s results[34]. Kimura, T. also showed that TNC accelerated adverse ventricular remodeling after MI by modulating macrophage polarization, we demonstrated once again that the up-regulation of TNC induces cardiac apoptosis and fibrosis[7].
- Lately, I suggest not reporting the figures in the discussion.
From your opinion, we have deleted the report of figures in discussion.
- At the end, I might add a comment. Data regarding the KO or OVER expression of METTL3 in reducing or increasing cardiac fibrosis is well established in the literature by other studies. It would have been better to cite them instead of reproducing data in this case. Nevertheless, I would ask you to perform experiments such as CC3/C3 and BAX/Bcl2, as well as the Caspase activities, to gain the strength of the research.
This is a great comment. We have finished all experiments you have suggested, such as CC3/C3 and BAX/BCL2, as well as the Caspase activities. Since we have carried out many studies on METTL3 in cardiovascular disease at the same time, we have confirmed the apoptosis effect of METTL3 in myocardial cells under hypoxia in relevant experiments. Since the article has not been officially published, we only provided relevant unpublished data here. Since this project was designed, we have only carried out the effects of METTL3 on myocardial infarction at the animal level, so almost all of our phenotype data are carried out at the animal level, and some mechanism experiments are carried out in cells.
(unpublished data)
- Wang, X., et al., Structural basis of N(6)-adenosine methylation by the METTL3-METTL14 complex. Nature, 2016. 534(7608): p. 575-8.
- Yoon, K.J., et al., Temporal Control of Mammalian Cortical Neurogenesis by m(6)A Methylation. Cell, 2017. 171(4): p. 877-889 e17.
- Dorn, L.E., et al., The N(6)-Methyladenosine mRNA Methylase METTL3 Controls Cardiac Homeostasis and Hypertrophy. Circulation, 2019. 139(4): p. 533-545.

Reviewer 3 Report
The manuscript titled " TNC accelerates hypoxia-induced cardiac injury in a METTL3- 2 dependent manner" identified m6A methyltransferase METTL3 as an important regulator that mediates the hypoxia-induced m6A modification on RNA transcripts to regulate the cardiomyocytes apoptosis and cardiac fibrosis. Overexpression of METTL3 promoted the hypoxia-induced TNC m6A modification to regulate the myocardial dysfunction. The work provided experimental evidence in support of the involvement of RNA epigenetics in the cellular response to hypoxia stress. The identification of METTL3 as a key regulator of hypoxia-induced pro-apoptosis and pro-fibrosis suggests a potential therapeutic approach for MI through targeting METTL3.
The manuscript is well-written.
Minor suggestion
1) Please include the anesthesia details in line 84.
Author Response
Thank you for your suggestion. We have added the anesthesia details. (lines 101-103).
The cardiac function of different groups of mice was evaluated with Vevo 770 instrument (Visual Sonics Inc., Toronto, Canada) under mild anesthesia. The mice were anesthetized with inhaled isoflurane (4.5% for induction of anesthesia and 1.5% for maintenance of anesthesia).
Reviewer 4 Report
Cheng et al., show that “TNC accelerates hypoxia-induced cardiac injury in a METTL3- 2 dependent manner”. The current study is interesting and relevant to the field of cardiovascular diseases. However, this reviewer has several concerns that need to be addressed.
1. Title of the manuscript seems inappropriate to the current reviewer. The author stated in the title “….hypoxia-induced cardiac injury…”, without any hypoxia induction in their present study.
2. The author may need to state the proper hypothesis of their study in the abstract section.
3. Figure 1c, 2g, 3g, 4c: The representative M-mode echocardiography images looks like they were not properly cropped from the original. It will be easier to understand by the reader if the original scales will be included with the images.
4. The graphs are different across the manuscript, some are dots plots, some are bar graphs, and some are dots plots with bar graphs. It would be good if the author could maintain a single format of the graphs across the manuscript. Dot plots with bar graphs would be appropriate.
5. Sample size for the experiment in Figures 2c, and 3d needs to be increased. Also, the author should state the proper sample size of each experiment in each figure legend.
6. It would be great if the author can measure the fibrosis in the tissue section from the different experimental groups.
7. The reviewer could not find the result or figure legends part for figures 5f and 5g in the manuscript. Also, the actinomycin D experiment that was stated in the result or figure legends section (lines 301-304, and line 321) did not correlate with the figure.
8. If the author can add a graphical abstract in the manuscript that would be great for the readers to understand the subject clearly.
Author Response
Thank you for your suggestions!
- Indeed, in this study, we are mainly engaged in animal experiments. We are used to simulating hypoxia-induced myocardial cell damage in acute myocardial infarction models. We accept your opinion and change the title to: TNC accelerates myocardial infarction in a METTL3-dependent manner
- We decided to rewrite the abstract section, and we added the proper hypothesis of our study.
Cardiac fibrosis and cardiomyocyte apoptosis are reparative processes after myocardial infarction(MI), which results in cardiac remodeling and heart failure at last. Tenascin-C (TNC) consists of four distinct domains, which is a large multimodular glycoprotein of the extracellular matrix. It is also a key regulator of proliferation and apoptosis in cardiomyocytes. As a significant m6A regulator, METTL3 binds m6A sites in mRNA to control its degradation, maturity, stability and translation. Whether METTL3 regulates the occurrence and development of myocardial infarction through the m6A modification of TNC mRNA deserves our study. Here, we have demonstrated that overexpression of METTL3 aggravated cardiac dysfunction and cardiac fibrosis after 4 weeks after MI. Moreover, we also demonstrated TNC resulted in cardiac fibrosis and cardiomyocytes apoptosis after MI. Mechanistically, METTL3 led to enhanced m6A level of TNC mRNA, and promoted TNC mRNA stability. Then, we mutated one m6A site “A” to “T”, the binding ability of METTL3 was reduced. In conclusion, METTL3 is involved in cardiac fibrosis and cardiomyocytes apoptosis by increasing m6A level of TNC mRNA and may be a promising target for the therapy of cardiac fibrosis after MI.
- It will be easier to understand by the reader if the original scales will be included with the images.
These pictures are cut from the original pictures. We provide some original pictures (including related original scales). However, in most of the literature, the images of mouse heart ultrasound will not provide scales[1-3]. I provided relevant literature images to support this .
(our original pictures)
( The novel butyrate derivative phenylalanine-butyramide protects from doxorubicin-induced cardiotoxicity)
( Ablation of lncRNA Miat attenuates pathological hypertrophy and heart failure)
( Cardiac dysfunction in Pkd1-deficient mice with phenotype rescue by galectin-3 knockout)
- We have adopted the best dot plots with bar graphs form for most images, and only some images that meet special conditions are in other formats.
- Sample size for the experiment in Figures 2c, and 3d needs to be increased. Also, the author should state the proper sample size of each experiment in each figure legend.
We have enlarged the sample size for the experiments in Fig 2c (n=6) and 3d (n=6). Also, we have stated all sample size of each experiment in each figure legend.
- It would be great if the author can measure the fibrosis in the tissue section from the different experimental groups.
We have added the measurement of fibrosis in the tissue section in Fig 1i, 2m, 3m.
- The reviewer could not find the result or figure legends part for figures 5f and 5g in the manuscript. Also, the actinomycin D experiment that was stated in the result or figure legends section (lines 301-304, and line 321) did not correlate with the figure.
We really lost some of the legends in this part. Now we have edited the legends in this part. The complete version is as follows: f. Actinomycin D was used to demonstrate that the stability of TNC mRNA was regulated by METTL3 in HL1 cells (n=5); g. Actinomycin D was used to demonstrate that the stability of TNC mRNA was regulated by METTL3 in AC16 cells (n=5);
The actinomycin D experiment that was stated in the result in lines 327: ... that of the presence of METTL3 (Figure 5f and 5g).
- If the author can add a graphical abstract in the manuscript that would be great for the readers to understand the subject clearly.
We have added a graphic abstract in the manuscript. We think it is better for readers to understand the subject.
(graphical abstract)
- Russo, M., et al., The novel butyrate derivative phenylalanine-butyramide protects from doxorubicin-induced cardiotoxicity. Eur J Heart Fail, 2019. 21(4): p. 519-528.
- Yang, L., et al., Ablation of lncRNA Miat attenuates pathological hypertrophy and heart failure. Theranostics, 2021. 11(16): p. 7995-8007.
- Balbo, B.E., et al., Cardiac dysfunction in Pkd1-deficient mice with phenotype rescue by galectin-3 knockout. Kidney Int, 2016. 90(3): p. 580-97.

Round 2
Reviewer 2 Report
Dear Authors,
Thank you for making changes to improve the work's scientific impact.
Author Response
Thanks a lot!